# Genetic Therapy for Intervertebral Disc Degeneration

**DOI:** 10.3390/ijms22041579

**Published:** 2021-02-04

**Authors:** Eun Ji Roh, Anjani Darai, Jae Won Kyung, Hyemin Choi, Su Yeon Kwon, Basanta Bhujel, Kyoung Tae Kim, Inbo Han

**Affiliations:** 1Department of Neurosurgery, CHA Bundang Medical Center, CHA University School of Medicine, Seongnam-si 13496, Korea; morolro@naver.com (E.J.R.); anjanianji09@gmail.com (A.D.); kyungjaewon88@gmail.com (J.W.K.); littlechoi88@gmail.com (H.C.); syunkwon@naver.com (S.Y.K.); basantabhujel86@gmail.com (B.B.); 2Department of Biomedical Science, CHA Bundang Medical Center, CHA University School of Medicine, Seongnam-si 13496, Korea; 3School of Medicine, Department of Neurosurgery, Kyungpook National University, Daegu 41944, Korea; nskimkt7@gmail.com; 4Department of Neurosurgery, Kyungpook National University Hospital, Daegu 41944, Korea

**Keywords:** genetic therapy, intervertebral disc degeneration, RNAi, mTOR signaling, CRISPR-Cas9, vector

## Abstract

Intervertebral disc (IVD) degeneration can cause chronic lower back pain (LBP), leading to disability. Despite significant advances in the treatment of discogenic LBP, the limitations of current treatments have sparked interest in biological approaches, including growth factor and stem cell injection, as new treatment options for patients with chronic LBP due to IVD degeneration (IVDD). Gene therapy represents exciting new possibilities for IVDD treatment, but treatment is still in its infancy. Literature searches were conducted using PubMed and Google Scholar to provide an overview of the principles and current state of gene therapy for IVDD. Gene transfer to degenerated disc cells in vitro and in animal models is reviewed. In addition, this review describes the use of gene silencing by RNA interference (RNAi) and gene editing by the clustered regularly interspaced short palindromic repeats (CRISPR) system, as well as the mammalian target of rapamycin (mTOR) signaling in vitro and in animal models. Significant technological advances in recent years have opened the door to a new generation of intradiscal gene therapy for the treatment of chronic discogenic LBP.

## 1. Introduction

### 1.1. Basic Anatomy of Intervertebral Disc

The intervertebral disc (IVD) is made of fibrous cartilage and is the most important functional part of the spine, presenting in between the vertebral bones providing flexibility to the vertebrae. It consists of the nucleus pulposus (NP), surrounded by the annulus fibrosus (AF), and the cartilaginous endplate [1,2,3,4,5,6,7]. NP cells are similar to chondrocytes, which consist of collagen fibers embedded with the gel-like aggrecan. AF also consists of collagen fibers lying parallel with the lamellae, whereas the cartilaginous endplate is made up of a thin layer of hyaline. All these tissues being sandwiched together yields important functional properties to the disc [4,8,9]. IVD acts as a shock absorber, helps in the movement of the vertebrae, and holds the vertebrae together. IVD degeneration (IVDD) results in chronic lower back pain (LBP), which is now a global concern, and causes more dysfunction than any other medical situation [10].

### 1.2. Pathophysiology of IVD Degeneration

IVDD is the main cause of chronic LBP, which can lead to severe disabilities. It is a crucial factor in the degradation of the structure and function of the IVD. IVDD could alter the structure of the extracellular matrix (ECM), leading to the weakening of tissues and changes in the cells. These changes produce a functional effect on the IVD and ECM composition. This is a crucial factor in the breaking down of the structure and function of the IVD [1,2,4,8]. The main reasons associated with IVDD include nutritional, environmental, and genetic factors [11,12]. These factors lead to changes in cell morphology, inflammation, an increase in senescence cells, apoptosis, and autophagy [13,14,15,16,17,18]. IVDD is characterized by a loss of NP cells (more rounded chondrocyte-like cells) and their replacement by cells with a fibroblast-like phenotype [19]. Inflammatory responses exacerbated by pro-inflammatory cytokines, such as tumor necrosis factor-α (TNF‑α) and interleukin-1β (IL‑1β), are known as major events that occur during IVDD and associated chronic LBP [20]. Senescent cells secrete a senescence-associated secretory phenotype (SASP), which increases the secretion of pro-inflammatory cytokines, chemokines, and tissue-damaging proteases. Increased amounts of senescent cells have been found in the degenerated disc [21]. Moreover, apoptosis and autophagy are known for two patterns of programmed cell death and play important roles in IVDD [22]. In a study, change by age or by heredity shows significant role for the defect in endplate of disc in association with IVDD [23]. IVDD is usually associated with high catabolism and less anabolism in the ECM. One of the main causes of IVDD is a change in the nutrition levels in the discs, which progresses in the presence of a reduced supply of oxygen and low pH; conditions that are harmful to the ECM of the IVD [16]. The possible environmental risk factors for IVDD are smoking, lack of exercise, an unsanitary lifestyle, trauma, and mechanical load. Moreover, with aging, an alteration in the disc ECM reduces the aggrecan level in the NP of the disc, and a decrease in the level of hydration along with functional changes of the matrix also occur. As a result, dehydrated NPs are unable to properly balance the forces in adjacent vertebrae. Thus, the forces eventually become dispersed in the AF region leading to mechanical and gradual structural destruction, which results in a radial annular tear and herniation of the NP region [15].

Some macroscopic changes include annulus tear, decreased disc hydration, loss of disc height, lamella disorganization of the annulus, and osteophytes formation [24,25]. These are generally common in the composition of the ECM. Molecular changes include alterations in genes, protein expression, matrix synthesis, catabolic mechanisms, growth factors, and cytokines [26]. Matrix changes are related to the anabolic mechanism; collagens and proteoglycans (PG) are the two main matrices involved in the degeneration process. A reduction in PG content in the NP is the starting process for the deterioration of the IVD [15]. Because the disc is structurally devoid of blood vessels, it is a hypoxic environment in which oxygen and nutrients are supplied through capillaries in the endplate. Therefore, as disc degeneration progresses, inflammation leads to more hypoxic and acidic conditions [4]. However, NP cells must proliferate and differentiate with a low metabolism, so the catabolic activity must be downregulated to maintain balance. Gene therapy, growth factors, and cell therapy could promote matrix formation. To induce disc regeneration by anabolism, high energy is required for matrix formation. On the other hand, with less energy and resources, different approaches, such as RNA interference (RNAi), can be used to promote disc regeneration. This helps to maintain homeostasis by upregulating anabolism and downregulating catabolism [27] (Figure 1).

### 1.3. Discogenic Low Back Pain (LBP)

Discogenic LBP is a multifactorial and complex disease and is the most common type of chronic LBP, accounting for 39% of cases. Discogenic LBP is caused by an inflammatory response during IVDD [28]. The increase in pro-inflammatory cytokines produced by degenerated and senescent disc cells has been considered as a nociceptive and noxious trigger of the painful condition [29].

### 1.4. Current Treatments for Chronic LBP due to IVD Degeneration

Current treatments for chronic LBP caused by IVDD include physiotherapy, medication, and surgery [30,31]. If there is no response to conservative treatments, surgery can be done. Surgery involves partial resection of the herniated disc that compresses the nerve roots and a fusion surgery, in which the intervertebral cage is inserted after the disc is completely removed [32,33]. However, many patients suffer from post spinal surgery syndrome (PSSS), that is, persistent or recurrent LBP after surgery [1,4]. The prevalence of PSSS has been reported to be approximately 30% [34,35]. There is a need for the development of new biological treatments that can be clinically applied for the purpose of inhibiting underlying disc degeneration, protecting degenerated discs, or regenerating the discs. This may include injection of various growth factors with or without carriers, use of cells with or without scaffolds, and genetic modification of gene expression through gene therapy. To date, these treatments have been studied in an animal model, but there are no biological agents for IVD regeneration [27,36].

## 2. Search Strategy

We searched the keywords of “Intervertebral disc”, “Gene therapy”, “RNA interference”, “CRISPR”, “autophagy”, and “mTOR signaling” in PubMed (https://pubmed.ncbi.nlm.nih.gov/) and Google Scholar from January 1997 and December 2020 and reviewed full-text articles (CRISPR: gene editing by the clustered regularly interspaced short palindromic repeats; mTOR: mammalian target of rapamycin).

## 3. Biological Approaches

As there are still many patients with chronic discogenic LBP who require refractory to conservative (medication and physiotherapy) and surgical treatments, there is an increasing unmet need for biological approaches including growth factor injection, cell-based therapy, tissue engineering, and gene therapy [27].

### 3.1. Growth Factor Injection

This is a method in which growth factors are injected directly into the degenerated disc by inducing matrix synthesis and preventing inflammation in the degenerated disc. A representative example is growth differentiation factor (GDF)-5, which plays an important role in the formation of the core of bones and joints, and in cell growth and differentiation in embryonic development [9,37]. One study showed the recovery effect of GDF-5 injection in the rabbit disc degeneration model [38]. However, when only the growth factor itself is injected, the long-term persistence within the disc is significantly lower [39]. In order to overcome these shortcomings, treatments such as gene therapy and tissue engineering are gradually being developed. In a recent study, GDF-5 injection was performed through tissue engineering in the form of injecting scaffolds or stem cells in a rat model [40].

In addition, numerous growth factors, including bone morphogenetic proteins (BMPs), insulin-like growth factor-1 (IGF-1), and transforming growth factor-β (TGF-β), have been reported to have therapeutic effects that slow or reverse IVDD [15]. In cartilage, TGF-β has been shown to participate in processes including cartilage formation, metalloproteinase production, and inflammatory responses. Gene expression of TGF-β was found to be promoted in osteoarthritis cells similar to those observed in degenerative disc cells. TGF-β and TGF-β type II receptors are generally present in herniated disc cells and in specimens of non-herniated human discs [15,41,42].

### 3.2. Cell Therapy

Among the various biological approaches, cell-based therapy appears to be the most promising technology for the treatment of IVDD. In this process, mesenchymal stem cells (MSCs) isolated from bone marrow, adipose tissue, umbilical cord Wharton’s jelly, and synovial membrane have been used due to their self-renewal capacity, anti-inflammatory properties, and their ability to regenerate degenerated disc cells [43,44,45]. The mechanism of stem cell therapy has been reported to have a paracrine effect (anti-inflammatory, anti-apoptotic), be capable of the restoration of degenerated discs, and initiate differentiation of implanted stem cells into NP cells [3,4,5,6,7,8,46].

### 3.3. Tissue Engineering

Cell therapy has shown some limitations in the repair of IVDD during studies in vitro, in vivo, and in many clinical trials [47]. From this point of view, the tissue engineering approach (the combination of growth factors, stem cells, and a scaffold) is more important because of the positive results of using different types of functional polymers such as chitosan, collagen, gelatin, alginate, hyaluronic acid, polyethylene glycol, polyurethane, polylactic acid, and polyglycolic acid. Based on cells and scaffolds, this strategy is an effective treatment for IVDD [48]. Because of its advantages, many researchers have developed various tissue engineering techniques for the replacement of degenerated discs in animal models [49,50,51]. For mild and moderate IVDD, stem cell transplantation may be an effective treatment. However, for severe IVDD that requires structural support, multifunctional treatment is possible through tissue engineering [4,5,6,7,8,46].

## 4. Gene Therapy

IVDD involves significant genetic risk factors. It has been reported that an imbalance in the synthesis and catabolism of certain important ECM components can be alleviated by transferring genes to IVD cells that encode factors that regulate matrix synthesis and catabolism [52]. Moreover, successful gene transfer to target cells within IVD in clinically relevant animal models of IVDD represents exciting new possibilities for the treatment of IVDD [52]. Here, we briefly summarize gene transfer to disc cells in in vitro and animal models, gene silencing via RNAi, gene editing via the clustered regularly interspaced short palindromic repeats (CRISPR) system, and the mammalian target of rapamycin (mTOR) signaling [53,54] (Table 1).

### 4.1. Gene Transfer to Target Disc Cells In Vitro andIn Vivo

One of the technologies required for gene therapy is a method of delivering the desired gene to the target. As an essential vehicle, a vector facilitates translocation into the nucleus along with the expression of the transgene. This is useful for transferring the gene of interest, i.e., complementary DNA (cDNA), a reverse-transcribed DNA molecule of the mRNA, to the host cells. Viral vectors have been widely used because they can efficiently rearrange their genetic material; however, non-viral mediated gene transfer to disc cells has been developed because of the potential risks associated with viral gene therapy [2,27].

#### 4.1.1. Virus Vector-Mediated Gene Transfer to Disc Cells

##### Retrovirus

In 1997, Wehling et al. [55] first introduced the idea of using in vitro gene transfer to reverse IVDD and successfully transfected the target gene into the in vitro-cultured chondrocyte cells isolated from bovine coccygeal vertebral endplates, using a retrovirus vector. The bacterial β-galactosidase (lacZ) gene and the cDNA of the human interleukin-1 (IL-1) receptor antagonist, were introduced into the chondrocyte cells. Transfer of the β-galactosidase (lacZ) gene to cultured cells produced approximately 1% lacZ positive cells, and transfer of the IL-1 receptor antagonist cDNA produced 24 ng/mL/10^6^ cells of the IL-1 receptor antagonist protein. This study showed the potential of ex vivo gene therapy for the degenerated disc [2].

##### Adenovirus

In 1998, Nishida et al. [56] reported the transfer of the lacZ gene to the disc in both in vitro and in vivo rabbit models using an adenovirus vector. This report was considered to be the first example of gene therapy that targeted NP cells in vitro and in vivo. For the in vitro model, disc NP cells isolated from female New Zealand white rabbits were cultured and transfected with an adenovirus construct encoding the lacZ gene. For the in vivo model, an adenovirus construct encoding the lacZ gene was injected directly into the NP of the rabbit’s lumbar IVD. X-Gal (5-bromo-4-chloro-3-indolyl-β-D-galactopyranoside) staining in vitro and in vivo showed the efficient transduction of disc NP cells, and reporter gene expression in vivo lasted for at least 12 weeks. This study showed that the exogenous gene was successfully delivered to the disc via an adenovirus vector in vitro and in vivo, demonstrating the potential of direct gene therapy for IVDD treatment [57,58].

##### Adeno-Associated Virus (AAV)

The activatorprotein-2α (AP-2α) has been reported to be involved in IVDD by regulating the expression of TGF-β1 and Smad3 [59]. AP-2α and TGF-β1 were upregulated in NP tissues of patients and rats with IVDD. Injection of low-expressing AP-2α and high-expressing TGF-β1 adeno-associated viral vectors into the IVD of rats reduced the expression of matrix metalloproteinase (MMP)-2, MMP-9, and Smad3 in NP tissue and increased the expression of aggrecan and Col-2. This study confirmed that knockdown of AP-2α restricted the expression of TGF-β1 and Smad3 to promote proliferation and reduce the senescence and apoptosis of NP cells in rats with IVDD [59]. AAVs could be effective tools for gene therapy [60,61].

##### Baculovirus

Liu et al. [62] treated rabbit IVD cells with baculovirus carrying the green fluorescence protein gene (Ac-CMV-GFP). As a result of confirming the expression of GFP in NP tissue by injection into the intervertebral discs of rabbits on days 7, 13, 20, and 28, Ac-CMV-GFP was expressed for a long time without toxicity to cells, showing the highest transduction rate, and GFP expression level at 7 days after transduction. This study indicates that baculovirus with exogenous genes can be safely used with high efficiency in rabbit NP cells, suggesting that it is a useful tool as a gene therapy vector for IVDD.

##### Lentivirus

In 2020, Zhao et al. [63] injected lentiviral vectors (LV-MMP3-shRNA and/or LV-Sox9) into rabbit lumbar discs to confirm that lentivirus-mediated MMP3 knockdown could attenuate IVDD. Magnetic resonance imaging (MRI) scans after 8, 12, and 24 weeks showed that IVDD was observed in animals injected with phosphate-buffered saline or 10^7 viral particles of the control virus. In contrast, IVDD was suppressed by 10^7 viral particles of LV-MMP3-shRNA or LV-Sox9. In addition, MMP3 knockdown or Sox9 overexpression stimulated collagen type II, aggrecan, and proteoglycan synthesis. In addition, the injection of a cocktail of LV-MMP3-shRNA and LV-Sox9 (5 × 10^6^viral particles, respectively) significantly delayed the development of IVDD and induced the production of the highest levels of collagen type II and proteoglycan. This study suggests that gene therapy targeting MMP3 is an effective way to delay IVDD in NP tissue.

#### 4.1.2. Non-Virus Vector-Mediated Gene Transfer to Disc Cells

##### RNA Interference (RNAi)

In 2020, Bi et al. [64] attenuated H₂O₂ induced acute inflammation by overexpressing the Klotho gene using RNAi using the IVDD rat model and inhibiting Toll-like receptor 4 (TLR4). As a result, Klotho inhibition and elevated TLR4 levels showed pro-inflammatory nuclear factor kappa B (NF-κB) signaling and cytokine expression in NP cells of all animal groups. When TLR4 overexpression reduced Klotho expression and interfered with TLR4 expression, the inhibitory effect of H₂O₂ was reduced in NP cells. Klotho knockdown by RNAi reduced the anti-inflammatory and IVD protective effects of the IVD degeneration model. Therefore, this study suggests that the expression of Klotho regulates TLR4-NF-κB signaling transduction [2,65,66].

##### Ultrasound Targeted Microbubble Destruction

In 2006, Nishida et al. [67] injected plasmid DNA encoding Green Fluorescence Protein (GFP)and luciferase into the IVD in a rat model. After being observed for 1, 3, 6, 12, and 24 weeks, the results showed that expression of the GFP transgene—visualized using ultrasonic target microbubble destruction—was clearly observed in NP cells. Luciferase analysis also revealed an 11-fold increase in luciferase activity in this group compared to the plasmid DNA group. In this study, using the ultrasonic-targeted microbubble destruction transfection method significantly improved the transfection efficiency of plasmid DNA in NP cells, which retained gene expression for up to 24 weeks.

##### Polyplex Micelle

In 2020, Huang et al. [68] observed the effect of mixed cationic block copolymers (MCBC) PNIPAM-b-PAsp (DET) and PEG-b-PAsp (DET) polyplex micelles using an miRNA-25-3p vector as the therapeutic agent in an IVDD rat model. It has been observed that IL-1β, ZIP8, and MTF1 increased and miRNA-25-3p decreased in degenerated tissue compared to normal tissue. The treatment of miRNA-25-3p inhibited ECM degrading enzyme expression and restored ECM protein expression. MCBC was able to effectively deliver miRNA-25-3p to NP cells, which delayed the progression of IVDD. In this study, polyplex micelles made from a vector that delivers miRNA-25-3p could be applied as a therapeutic agent [2,69,70,71,72].

To date, approximately 40% of adenovirus and retroviral vectors are used in clinical trials. However, research on non-viral vectors that can replace viral vectors is also underway due to various side effects and risks associated with viral vectors. Research and development to overcome the problems of viral vectors and nonviral vectors is actively being conducted, and many studies are being conducted. For the most promising gene therapy in terms of the structural properties and function of IVD cells, it is important to invasively target only IVD cells. Vectors that have been developed to date that are required for gene targeting still need further development. If such challenges can be overcome, safe and efficient clinical applications in IVDD treatment will be possible.

### 4.2. Clustered Regulatory Interspaced Short Palindromic Repeats-Associated Cas9 (CRISPR/Cas9)

CRISPR/Cas9 is a convenient and versatile tool for modifying the genome. This is one of the more accurate and efficient treatments and is easier to use compared to other genome editing technologies [73]. The working principle of CRISPR-Cas9 is detailed in Figure 2 [74]. In 1987, Ishino et al. originally discovered CRISPR and subsequently observed CRISPR-associated Cas gene proteins. Later, the CRISPR/Cas9 editing system was first described by Cong et al. in 2013 [75]. The system has been used in both viral and non-viral delivery methods [76]. For the gene therapy strategy of IVDD, the regulation of cytokine receptors, TNF receptor 1(TNFR1) and IL-1 receptor 1 (IL1R1) signaling by a lentiviral CRISPR epigenome editing system was tested in human IVD cells to downregulate TNFR1 and IL1R1 pro-inflammatory signals [2,69]. They also showed that TNFR1 expression was downregulated by increasing aggrecan and decreasing MMP3 levels in TGFR1 genome editing. However, in the case of IL1R1 genome editing, IL1R1 expression was not downregulated and did not show any changes in aggrecan and MMP3 [77,78]. Thus the ability of the CRISPR/Cas9 genome editing system can be demonstrated by regulation of TNFR1 but not by IL1R1. This shows that regulation of these genes requires a finite perspective [2,77,78].

Similarly, Cambria et al. [79] reported that the transient receptor potential vanilloid type 4 (TRPV4) gene was successfully knocked out in vitro by the CRISPR/Cas9 gene-editing system in AF-injured patients with chronic LBP. By using CRISPR, they investigated the role of the TNFR4 gene in the regulation of IL6 and IL8 using CRISPR and the reduction in inflammation induced by hyper physiological stretching of AF cells. They also suggested that in the future studies, this could be used for targeting other genes for treatment of IVDD. Stover et al. [80] investigated changes in dorsal root ganglion (DRG) neuron activity in a rat model and demonstrated a CRISPR epigenetic editing system for pain-based neuromodulatory therapy in IVDD. They used the lentiviral CRISPR epigenome editing vectors to target the AKAP150 gene in peripheral neurons and demonstrated the effects of target promoter histone methylation on gene expression. After delivery to DRG neurons, epigenome editing vectors induced increased neuronal activity [69,80]. Krupkova et al. suggested that CRISPR/Cas9 can be applied to gene knockout, gene editing, inhibition, and gene expression activation, suggesting that these methods could be promising new tools that may play a role in IVDD treatment [53,81].

### 4.3. Correlation between IVD Degeneration and mTOR Signaling

mTOR is a serine/threonine protein kinase, considered as a necessary mediator for cellular metabolism, which determines whether catabolic and anabolic effects occur depending on nutritional levels. The ability to regulate anabolism and catabolism in vivo may reduce IVDD and increase disc regeneration [82]. mTOR contains serine and threonine proteins, which are necessary for processes related to mammalian survival, such as cell growth, proliferation, motility, survival, and autophagy.

Studies have shown that mTOR signaling regulates cells during metabolic processes. It induces anti-apoptosis, proautophagy, anti-matrix catabolism, and anti-senescence in disc cells. Studies on the Akt-dependent effect of inhibition of mTOR complex 1 (mTORC1) on mTOR have confirmed that mTOR signaling plays an important regulatory role in disc cells [3,83]. In addition, the roles and involvement of autophagy and mTOR signaling in IVDD were confirmed through both in vitro and in vivo studies using human and rabbit disc cells, with increased autophagy, increased disc cell death, apoptosis, metabolism, and increased aging also being noted [54]. Cell loss is a major feature of IVDD and is caused by cell apoptosis. In humans and rodents, the incidence of cell apoptosis is high with disc aging or degeneration [84]. Based on the results of the study, autophagy and mTOR signaling have been demonstrated to be used to combat harsh disc environments such as low glucose, low oxygen, acidic pH, and limited nutrient availability under an inflammatory milieu [83,85]. Selective RNAi-mediated and pharmacological inhibition of mTORC1 protects against inflammation-induced apoptosis, senescence, and ECM catabolism in NP cells [54]. In addition, studies have shown protective effects of rapamycin (mTORC1 inhibitor) against inflammation-induced apoptosis and catabolic gene expression in human chondrocytes [86]. Activation of the phosphatidylinositol 3-kinase (PI3K)/Akt pathway protects against IVDD through multiple mechanisms, including down regulation of MMP-3 and MMP-13 expression and upregulation of Sox9 expression, reduction in caspase-3 activity, and inhibition of apoptosis through activating mTOR [87]. It has been reported that resveratrol—in combination with 17β-estradiol—increases levels of activated phosphorylated mTOR (P-mTOR) and phosphorylated glycogen synthase kinase-3β (P-GSK-3β) leading to the down-regulation of caspase-3, inhibition of IL-1β-induced NP cell apoptosis and recovery of cell viability [88].

The importance of mTOR in IVDD has recently received a great deal of attention due to the high levels of mTOR signaling molecule expression and phosphorylation in human intermediate degenerated discs [54]. Therefore, an accurate understanding of the underlying molecular mechanisms is important to directly guide the development of biological therapies that target the mTOR signaling pathway for treating IVDD.

## 5. Future Perspectives

IVDD is a multifactorial disease involving the occurrence of both environmental and genetic effects at the same time. The mechanisms of IVDD have been clarified to some extent, but they have not been completely elucidated. Recent advances in science and research have improved our understanding of the mechanisms involved. Genetic mutations of genes involved in matrix formation could affect disc function or biochemical processes, which could lead to IVDD. Gene therapy, as a promising field of study, has the potential to exert therapeutic effects on the treatment of IVDD.

For gene therapy, gene editing technology CRISPR-Cas9 is an innovative technology in the scientific and medical fields that can modify target genes, which could potentially lead to accurate and efficient therapeutic effects. However, for gene editing technology, there are many concerns related to its clinical application, such as off-target effects, low editing efficiency, and immunogenicity. Gene editing technology can be clinically applied if off-target effects can be resolved. Immunogenicity and mutational variability are problems that arise when using CRISPR-Cas9 with viral vectors. If these technologies are to be applied to treat diseases, safer non-viral vectors need to be optimized and improved. In order for CRISPR-Cas9 to be applied for clinical use, it is necessary to focus on solving the problems presented. In addition, new research efforts are essential for enabling future clinical applications [74,88]. mTOR signaling is associated with IVDD in animal models. With this in mind, future therapeutic strategies for IVD regeneration based on gene therapy could target the destruction of mTORC1 and Regulatory-associated protein with mTORRAPTOR. Safety, high cost, and transfection are challenges that must be overcome for the development of IVDD treatments.

A direction for future studies could be to focus specifically on prophylactic and long-term regenerative treatments. Additionally, it is important to investigate more genetic defects. Advances in molecular and cellular biology, including genetic research, have enabled targeted therapies, and studies of intracellular signaling pathways need to move in the direction of providing targeted therapeutic pathways. More innovative technologies will be needed to overcome the obstacles encountered in gene therapy. As research related to gene therapy grows and gradually expands, IVDD may effectively overcome be in the future [2,9,89].

## Figures and Tables

**Figure 1 ijms-22-01579-f001:**
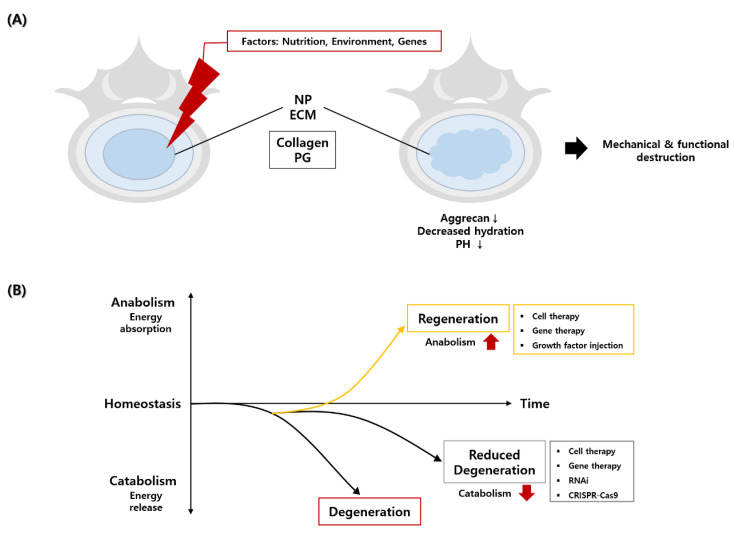
A schematic diagram of the cause of intervertebral disc degeneration and strategies to reduce disc degeneration and promote disc regeneration. (**A**) Nutritional, environmental, and genetic factors have caused changes in collagen and proteoglycan (PG) and the composition of the extracellular matrix (ECM) in nucleus pulposus (NP), resulting in a decrease in aggrecan and pH and loss of hydration, which systematically and functionally destroyed NP. (**B**) Regeneration of degenerated disc could be induced by an increase in anabolism and a decrease in disc degeneration could be induced by a decrease in catabolism.

**Figure 2 ijms-22-01579-f002:**
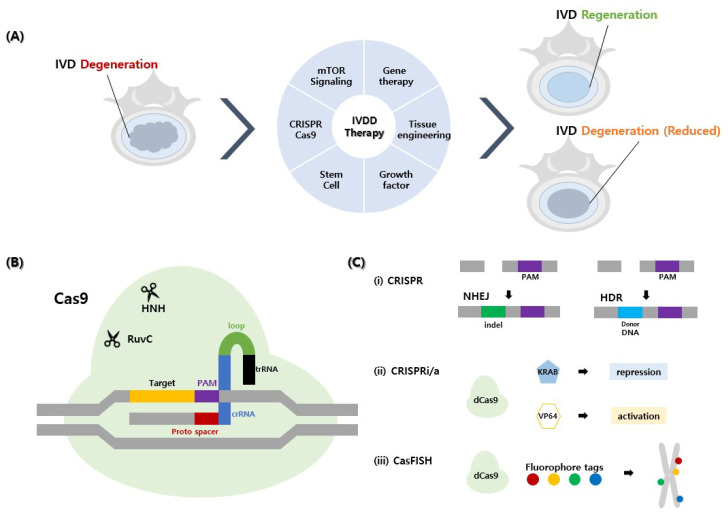
Schematic illustration of novel biological approaches and CRISPR-Cas9 as a gene therapy for the treatment of intervertebral disc degeneration (IVDD). Representative newly discovered biological approaches for the treatment of IVDD are shown as follows. (**A**) A simple diagram illustrating regeneration of IVDD using gene therapy, tissue engineering, growth factor, stem cell, CRISPR-Cas9, and mTOR signaling. (**B**) The form in which sgRNA, Cas9, and DNA are attached is called CRISPR-Cas9. sgRNA consisting of proto-spacer, crRNA, loop, and trRNA is cut and attached to the DNA proto spacer adjacent motif (PAM) sequence through Cas9 using HNH and Ruvc scissors. (**C**) The recovery mechanism of CRISPR-Cas9 in the intervertebral disc is as follows: (i) non-homologous end joining (NHEJ) of CRISPR inserts or deletes indels, and homology-directed repair (HDR) inserts new segments of DNA. (ii) CRISPR interference (CRISPRi) and CRISPR activation (CRISPRa) use dead Cas9 (dCas9) as a technology to mediate transcription site gene expression but have no cleavage effect and guide-target DNA at the start site of transcription. When combined with the KRAB domain, it inhibits transcription, and when combined with VP64, it activates the transcription of target DNA. (iii) Cas9-mediated fluorescence in situ hybridization (CASFISH) is dCas9, which enables DNA observation using fluorophore tags (multicolor). CRISPR: clustered regularly interspaced short palindromic repeats.

**Table 1 ijms-22-01579-t001:** Recent studies of gene therapy associated with intervertebral disc degeneration.

Gene Therapy		Target Gene	Target Cell	Reference
Viral vector	Retro	Bacterial lacZ, Human IL-1 receptor antagonist	Bovine chondrocytic cell	[55]
	Adeno	Bacterial lacZ	RabbitdiscNP cell	[56]
	Adeno-associated	AP-2α, TGF-β1	RatdiscNP cell	[59]
	Baculo	GFP	RabbitdiscNP cell	[62]
	Lenti	MMP3, Sox9	RabbitdiscNP cell	[63]
Nonviral Vector	RNAi	Klotho, TLR4	RatdiscNP cell	[64]
	UTMD	GFP, Firefly luciferase	RatdiscNP cell	[67]
	Polyplex micelle	miRNA-25-3p, Firefly luciferase	HumanandRatdiscNP cell	[68]
CRISPR	Cas9	TRPV4	HumandiscAF cell	[79]
mTOR pathway	PI3K/AKT	GSK-3β, NF-kappaB, caspase3, mTOR	RatdiscNP cell	[87]

Abbreviations: AF, annulus fibrosus; AP, activator protein; GFP, green fluorescence protein; IL, interleukin; NP, nucleus pulposus; miRNA, MicroRNA; MMP, matrixmetalloproteinase; mTOR, mammalian target of rapamycin; PI3K, phosphoinositide 3-kinase; Sox, SRY-box transcription factor; TGF, Transforming Growth Factor; TLR, Toll-like receptor; TRP, transient receptor potential; UTMD, ultrasound targeted microbubble destruction.

## Data Availability

Not applicable.

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
