# Peer review of "Genetic Therapy for Intervertebral Disc Degeneration"

_ijms, 2021, doi:10.3390/ijms22041579_

Round 1

Reviewer 1 Report

The manuscript entitled “Genetic therapy for intervertebral disc degeneration” describes the physiopathology of intervertebral disc degeneration and summarizes novel approaches to treat this disease which so far has not been yet fully explored.

Although the novelty and interesting treatment approaches described, there are several comments that should be addressed by the authors.

Major comments:

The major concern throughout the paper is the number of references included. The authors, lack to include reference papers in key sentences when describing the causes of the disease and treatment approaches (e.g. line 62-63 changes in cell morphology, inflammation, senescence and apoptosis). I suggest the authors to review this point and to include the reference papers.

Manuscript should be re-organized. Authors should include an introduction (IVD anatomy, pathophysiology of the disease and current therapeutic strategies) and a second section by highlighting novel approaches to treat IVD (from section 1.5 to section 4). Final section for future perspectives should be included as well.

In line with the previous comment, the authors give so many details related to the novel approaches mainly focusing on the description of the technique rather than explaining the impact of the treatment. I suggest the authors to reduce technical information and to clearly exemplify (based on published data) how new approaches will improve IVD treatment.

I doubt in the application of CRISPR/Cas9 technology for IVD treatment. Please rewrite this section and include reference papers describing the use of CRISPR for this disease.

The importance of mTOR pathways in IVD degeneration should be explained better. The authors describe the pathways, however poor information has been included summarizing the importance of targeting mTOR for IVD degeneration.

Figure 2 should include all novel approaches to treat IVD.

Minor comments:

Grammar and spelling should be addressed throughout the manuscript. The authors have important grammar mistakes such as verbs, plurals and in some cases, they have sentences that make no sense or are general with no information (e.g. line 77-78; line 80; line 98; line 102; line 106-108).

Abstract should be adapted to the new version of the manuscript.

Altogether, the manuscript should be re-written and refocus according to the comments for publication in ijms.

Author Response

Reviewer 1 comments:

Major comments:

#1 The major concern throughout the paper is the number of references included. The authors, lack to include reference papers in key sentences when describing the causes of the disease and treatment approaches (e.g. line 62-63 changes in cell morphology, inflammation, senescence and apoptosis). I suggest the authors to review this point and to include the reference papers.

Response#1:  

Thank you for pointing this out. We included reference papers. (Edit line 54-62)

#2 Manuscript should be re-organized. Authors should include an introduction (IVD anatomy, pathophysiology of the disease and current therapeutic strategies) and a second section by highlighting novel approaches to treat IVD (from section 1.5 to section 4). Final section for future perspectives should be included as well.

Response#2:  

Thank you for pointing this out. We reorganized our manuscript.

#3 In line with the previous comment, the authors give so many details related to the novel approaches mainly focusing on the description of the technique rather than explaining the impact of the treatment. I suggest the authors to reduce technical information and to clearly exemplify (based on published data) how new approaches will improve IVD treatment.

Response#3:  

Thank you for pointing this out. We reduced technical information and add explanation of the impact of the treatment.

#4 I doubt in the application of CRISPR/Cas9 technology for IVD treatment. Please rewrite this section and include reference papers describing the use of CRISPR for this disease.

Response#4:  

Thank you for pointing this out. The explanation of CRISPR-Cas9 was shortened and the content was summarized. And we added the latest papers related to Intervertebral disc degeneration. (Edit line 256-283)

#5 The importance of mTOR pathways in IVD degeneration should be explained better. The authors describe the pathways, however poor information has been included summarizing the importance of targeting mTOR for IVD degeneration.

Response#5:  

Thank you for pointing this out. We have added contents related to IVD degeneration and rewritten recent research on mTOR signaling. (Edit line 300-343).

#6 Figure 2 should include all novel approaches to treat IVD.

Response#6:  

Thank you for pointing this out. We have added figure 2 (A) related to novel approaches. (Edit line 285)

Minor comments:

Grammar and spelling should be addressed throughout the manuscript. The authors have important grammar mistakes such as verbs, plurals and in some cases, they have sentences that make no sense or are general with no information (e.g. line 77-78; line 80; line 98; line 102; line 106-108). 

Abstract should be adapted to the new version of the manuscript.

Response:  

line 77-78: We are grateful thanks for reviewers. It has been corrected in the revised manuscript (Edit line 75-77).

line 80: It has been corrected in the whole sentence (Edit line 78-80).

line 98, 102: We deleted and rewritten it (Edit line 95-98).

line 106-108 We deleted and rewritten it (Edit line 100-111).

Reviewer 2 Report

1. The topic is interesting. The paper is a review of the current status of genetic therapy for IVD degeneration.The authors should add a "review flow diagram" to show how they selected those papers.

2. Line 32 "In order to improve the vectors that have been developed so far in addition to gene therapy, further research is needed", ok but what is the aim of this review, can you better describe it.

3. References must be improved, please look and add these references to your manuscript:

Ahsan MK et al. Lumbar revision microdiscectomy in patients with recurrent lumbar disc herniation: A single-center prospective series. Surg Neurol Int 2020;11:404. doi:10.25259/SNI_540_2020

Perrini P et al. Anterior cervical corpectomy for cervical spondylotic myelopathy: Reconstruction with expandable cylindrical cage versus iliac crest autograft. A retrospective study. Clin Neurol Neurosurg. 2015 Dec;139:258-63. doi: 10.1016/j.clineuro.2015.10.023.

4. Figure 2 appears interesting. 

Author Response

Reviewer 2 comments:

  1. The topic is interesting. The paper is a review of the current status of genetic therapy for IVD degeneration. The authors should add a "review flow diagram" to show how they selected those papers.

Response 1:  

Thank you for pointing this out. We have added Search Strategy.

(Edit line 112-115)

  1. Line 32 "In order to improve the vectors that have been developed so far in addition to gene therapy, further research is needed", ok but what is the aim of this review, can you better describe it.

Response 2:  

Thank you for pointing this out. We deleted and rewritten it (Edit line 20-32)

  1. References must be improved, please look and add these references to your manuscript:

Ahsan MK et al. Lumbar revision microdiscectomy in patients with recurrent lumbar disc herniation: A single-center prospective series. Surg Neurol Int 2020;11:404. doi:10.25259/SNI_540_2020

Perrini P et al. Anterior cervical corpectomy for cervical spondylotic myelopathy: Reconstruction with expandable cylindrical cage versus iliac crest autograft. A retrospective study. Clin Neurol Neurosurg. 2015 Dec;139:258-63. doi: 10.1016/j.clineuro.2015.10.023.

Response 3:  

 Thank you for pointing this out. We have added Search Strategy(Edit line 103, 104) .

  1. Figure 2 appears interesting. 

Response 4:  

We grateful thank you for your looking interesting.

Total revision part:

We have modified figure 1(B) and Table 1. In Figure 1 (B), the correlation between anabolism and catabolism of degeneration and regeneration was changed to make it easier to see at a glance. Unlike previous tables that only explained the characteristics of vector, recent researches of Gene therapy related to IVD degeneration were selected. (Edited line 285, 344-348)

Round 2

Reviewer 2 Report

Authors solved all criticisms and improved the paper. Well done.

Author Response

Thank you very much!